# *Wolbachia* Detection in Field-Collected Mosquitoes from Cameroon

**DOI:** 10.3390/insects12121133

**Published:** 2021-12-17

**Authors:** Roland Bamou, Adama Zan Diarra, Marie Paul Audrey Mayi, Borel Djiappi-Tchamen, Christophe Antonio-Nkondjio, Philippe Parola

**Affiliations:** 1Aix Marseille Univ, IRD, AP-HM, SSA, VITROME, 13005 Marseille, France; bamou2011@gmail.com (R.B.); adamazandiarra@gmail.com (A.Z.D.); 2IHU-Méditerranée Infection, 13005 Marseille, France; 3Vector Borne Diseases Laboratory of the Biology and Applied Ecology Research Unit (VBID-URBEA), Department of Animal Biology, Faculty of Science, University of Dschang, Dschang P.O. Box 067, Cameroon; mayimariepaulaudrey@yahoo.com (M.P.A.M.); borel_tchamen@yahoo.com (B.D.-T.); 4Institut de Recherche de Yaoundé (IRY), Organisation de Coordination pour la Lutte Contre les Endémies en Afrique Centrale (OCEAC), Yaoundé P.O. Box 288, Cameroon; antonio_nk@yahoo.fr

**Keywords:** *Wolbachia* infection, mosquito, phylogeny, *16s rRNA*, *23s rRNA*, Cameroon

## Abstract

**Simple Summary:**

*Wolbachia* bacteria from different strains, carried by many insects and nematodes, can interact in many ways with their hosts by changing their biology in different ways, including by suppressing vector population and reducing parasite transmission. Consequently, *Wolbachia* play an important role in vector control strategies. This study assessed the prevalence of natural *Wolbachia* infections in mosquitoes collected in Cameroon. Despite the low prevalence that was revealed, *Wolbachia* spp. were found in eight species of field-collected mosquitoes and are closely related to clades A and B. *Aedes aegypti* and *A. gambiae* sl., the main vectors of dengue and malaria, respectively, were not infected in this study, while *C. moucheti* recorded a high prevalence (46.67%). Future characterisation of the *Wolbachia* bacteria obtained is needed.

**Abstract:**

*Wolbachia* spp., known to be maternally inherited intracellular bacteria, are widespread among arthropods, including mosquitoes. Our study assessed the presence and prevalence of *Wolbachia* infection in wild mosquitoes collected in Cameroon, using the combination of *23s rRNA Anaplasmatacea* and *16s rRNA Wolbachia* genes. Mosquitoes that were positive for *Wolbachia* were sequenced for subsequent phylogenetic analysis. Out of a total of 1740 individual mosquitoes belonging to 22 species and five genera screened, 33 mosquitoes (1.87%) belonging to eight species (namely, *Aedes albopictus*, *A. contigus*, *Culex quinquefasciatus*, *C. perfuscus*, *C. wigglesworthi*, *C. duttoni*, *Anopheles paludis* and *Coquillettidia* sp.) were found to be positive for *Wolbachia* infections. *Wolbachia* spp. were absent in *A. gambiae* and *A. aegypti*, the main vectors of malaria and dengue, respectively. Phylogenetic analysis of the *16S RNA* sequences showed they belong mainly to two distinct subgroups (A and B). This study reports the presence of *Wolbachia* in about eight species of mosquitoes in Cameroon and suggests that future characterisation of the strains is needed.

## 1. Introduction

Mosquitoes (Diptera, Culicidae) are one of the most diverse groups of arthropods in the world and are found in a wide range of aquatic and terrestrial habitats with varying morphological and behavioural adaptations [1,2]. Their feeding behaviour as haematophagous insects gives them the ability to transmit a huge amount of pathogens, including viruses, bacteria, protozoa and nematodes from one vertebrate host to another [3]. In Cameroon, mosquitoes are implicated in the transmission of 26 arboviruses, malaria parasites and filarial worms causing diseases to humans, birds and great apes [4]. These diseases account for about 17% of infectious diseases in the world, and at least half of the world’s population lives in areas where mosquito-borne diseases are endemic [4]. It has been noted that mosquitoes could be a vector of bacterial infectious diseases to humans, although only *Rickettsia felis* has been detected to date [5,6]. To reduce the burden of these threats, mosquito control measures have been developed, mainly based on chemical control using long-lasting insecticidal nets and indoor residual spraying [7]. Despite the measurable success of these tools in reducing the malarial burden, resistance to mosquito insecticides has emerged and now limits the effectiveness of these tools, calling for the development of new control strategies.

The genus *Wolbachia* includes maternally inherited endosymbiotic bacteria that naturally infect disparate ranges of insects, including mosquitoes of medical and veterinary importance [8,9,10,11]. It has long been thought that *Wolbachia* spp. were absent in some mosquitoes, such as *A. aegypti* and *Anopheles gambiae* mosquitoes [12]. However, recent studies have detected *Wolbachia* in many genera of mosquitoes including *Aedes*, *Anopheles*, *Culex*, *Armigeres*, *Mansonia*, *Coquillettidia*, *Culiseta*, *Hodgesia*, *Ochlerotatus*, *Tripteroides* and *Uranotaenia* [10,11,13,14,15]. Recently, the work of Ayala et al. [16] in Central Africa (Gabon) revealed that *Wolbachia* is largely prevalent among diverse groups of Anopheles species. *Wolbachia* may be distributed horizontally [17,18] between mosquito populations and was also confirmed to be maternally transmitted within the *A. moucheti* population in Cameroon [10].

A recent study carried out in Italy highlighted the effect of this bacteria on the reduction in fertility in the arbovirus vector *A. albopictus* [19]. Experimental trials on the major malaria vector *A. gambiae* have also revealed the potential of *Wolbachia* to have an impact upon the replication of human malaria parasite *Plasmodium falciparum* [6,17,18]. Furthermore, *Wolbachia* have been reported to block the transmission of dengue, zika and yellow fever viruses [20,21,22,23,24].

Although these bacteria are highly prevalent (56%) in *A. moucheti* malarial vectors in Cameroon, their distribution among other mosquito groups remains poorly understood [10]. In addition, knowledge about their prevalence remains limited. Data regarding *Wolbachia* in mosquitoes in Cameroon is lacking, and this study provides an opportunity to identify new *Wolbachia* strains that could be characterised. New *Wolbachia* strains are interesting candidates for vector control as they may confer useful phenotypes when transinfected into alternative mosquito hosts. In this study, the current prevalence of *Wolbachia* in different mosquito species was determined, and their sequences and phylogenetic were subsequently investigated using *23S rRNA Anaplasmataceae* and *Wolbachia 16S rRNA* targeting 550 bp and 460 bp of the *16S* and *23S* genes, respectively.

We tested the hypothesis that different mosquito species from Cameroon may also display different levels of prevalence of *Wolbachia* endosymbiont infection.

## 2. Materials and Methods

### 2.1. Biological Samples

Field-collected mosquito samples were collected between 2017 and 2021 in different localities of Cameroon (Yaoundé, Dschang, Nyabessang) using different collection techniques, including the human landing catch (HLC), Centers for Disease Control light traps (CDC-LTs), sweep nets for adult collections and dipping for immature stages. CDC-LT collections were conducted both indoors and outdoors in four selected houses between 7 pm and 6 am for at least three consecutive nights per season. Traps were placed indoors near someone sleeping under a net at about 1.5 m from the ground and outdoor on the veranda. The same number of houses were randomly selected per collection site for HLCs. In each house, indoor and outdoor mosquito collections were carried out between 7 pm and 6 am by two teams of two people per house using mouth aspirators. The two teams (indoor/outdoor collectors) exchanged positions in each homestead every hour of the night. Mosquito collection bags were exchanged hourly. Immature mosquito stages at breeding sites (abandoned tyres, metal and plastic containers, gutters and stagnant water pools) were collected using the dipping technique, while sweep nets were used to catch adult mosquitoes resting on vegetation. The mosquitoes collected belonged to the genera *Anopheles*, *Culex*, *Aedes*, *Lutzia*, *Mansonia*, *Coquillettidia* and *Eretmapodites*. Each mosquito species was first identified morphologically using a binocular microscope and identification keys [25,26,27], and this was later confirmed by matrix-assisted laser desorption ionization–time of flight mass spectrometry (MALDI-TOF MS) and molecular analysis [28]. Damaged and unidentified specimens (due to the absence of diagnostic characteristics and the loss of legs for MALDI-TOF MS identification) were identified using molecular tools. Mosquitoes were stored either in silicate or in empty containers (Eppendorf tubes or petri dishes with no preserving product) at room temperature, or in silicate, RNAlater, 70% alcohol and in empty containers in the fridge at −20 °C for two to 48 months, depending on the time of collection.

Using a sterile surgical blade, mosquito legs were removed and analysed using MALDI-TOF MS for species identification. The rest of the mosquito body, and at times the whole mosquito, were used for molecular identification and microbial detection.

### 2.2. MALDI TOF MS Analyses and Molecular Identification of Mosquitoes

The identification of the mosquito species was confirmed by MALDI-TOF MS and molecular biology analyses, and these results have been presented elsewhere. Briefly, the legs of specimens were introduced to 1.5 mL Eppendorf tubes containing 20 µL of protein extraction mix (70% formic acid (*v*/*v*) and 50% acetonitrile (*v*/*v*)) and grinded using glass beads in a Tissue Lyser [29,30]. One microlitre of each protein extract was deposited on a target plate in quadruplicate, and one microlitre of matrix was added to each spot and allowed to dry. Protein mass spectra were obtained using a Microflex LT MALDI-TOF mass spectrometer (Bruker Daltonics, Bremen, Germany) [31] and crosswise tested against an in-house database available at VITROME, containing spectra of known mosquito species. The results are presented by the reliability of species identification, estimated as the log score value (LSV) and calculated using a biostatistical algorithm from the MALDI Biotyper software v.3.0. The LSV range from 0–3 and values equal to or greater than 1.8 are considered as threshold values for the species identification of mosquitoes. Species that were not identified through morphological and MALDI-TOF MS methods or with no agreement between the methods were further analysed by molecular biology using cytochrome oxidase subunit 1 for invertebrate *Cox 1* genes [28]. DNA was individually extracted from the carcasses (specimens without legs) of mosquito specimens (*n* = 1740) using the QIAamp DNA tissue extraction kit (Qiagen, Hilden, Germany), according to the manufacturer’s instructions.

### 2.3. Mosquito Microbiota Analysis

Mosquito samples were submitted to five bacteria screening groups, namely *Rickettsia* spp., *Borrelia* spp., *Bartonella* spp., *Coxiella burnetii* and *Anaplasmataceae*. Mosquitoes were pooled (4–10 individual mosquitoes) and screened for the detection of *Rickettsia* spp., *Borrelia* spp., *Bartonella* spp., *Coxiella burnetii* and Anaplasmataceae using quantitative PCR (qPCR) [32,33]. When a pool was found to be positive for one type of bacteria, specimens were then analysed individually using the same method. Target genes, primers and probe sequences (Table 1) were used as previously described in other studies [32,33]. DNA from laboratory-cultured strains of *Borrelia crocidurae*, *Rickettsia africae*, *Bartonella quintana*, *Coxiella burnetii* and *Ehrlichia canis* were used as positive controls. DNA free of pathogens from laboratory-reared ticks was used as a negative control. Samples were considered to be positive when the Ct < 36 for all bacteria tested. Later on, samples (from individual specimens) positive for Anaplasmataceae underwent standard PCR amplification and sequencing of the *23s rRNA* and *16s rRNA* genes, as described by Diarra et al. [32].

### 2.4. Sequencing and Phylogenetic Analyses

Sequencing and phylogenetic analyses of positive samples (33) was performed as described elsewhere [34] using the *16S rRNA* and *23S rRNA* genes targeting 550bp and 460bp of the *16S* and *23S* genes, respectively. The products obtained were visualised on 1.5% agarose gel stained with SYBR Safe and purified using a Macherey Nagel (NucleoFast 96 PCR, Düren, Germany) plate. Sequencing was performed using the BigDye Terminator v1.1, v3.1 5x Sequencing Buffer (Applied Biosystems, Warrington, UK) and run on an automated sequencer. The sequence chromatograms that were obtained were assembled and edited using Chromas Pro1.77 (Technelysium Pty Ltd., Tewantin, Australia). The sequences obtained were used to perform BLAST searches against the National Center for Biotechnology Information (NCBI) GenBank database and were then aligned using BioEdit. A phylogenetic tree was constructed and edited using the maximum likelihood method with the model selection determined by TOPALI v2.5 and MEGA11, respectively. Statistical support for internal branches of the trees was evaluated by bootstrapping with 1000 iterations.

## 3. Results

In total, about 1740 mosquitoes collected in the field were analysed, and their abundance varied with the species. Twenty-two species belonging to five genera were identified and confirmed by a molecular analysis tool or MALDI TOF MS. In this paper, we focus on the detection of mosquito-borne bacteria circulating in Cameroon.

From the five bacterial groups screened in this study (*Rickettsia* spp., *Borrelia* spp., *Bartonella* spp., *Coxiella burnetii* and Anaplasmataceae), only DNA particles of Anaplasmataceae were found in our samples (33/1740; 1.89%). Infected samples included *Anopheles* spp. (*A. paludis*), *Aedes* spp. (*A. albopictus*, *A. contigus*), *Culex* spp. (*C. quinquefasciatus*, *C. moucheti*, *C. wigglesworthi*, *C. perfuscus*), *Lutzia* spp. (*L. tigripes*) and *Coquillettidia* spp. The prevalence of infection according to mosquito species is presented in Table 2. Following sequencing, all positive samples (33) using the *23s rRNA* gene were found to be positive for *Wolbachia* spp. (*W. pipientis*, *Wolbachia* endosymbiont of *Ctenocephalides felis* wCfeT, *Wolbachia* endosymbiont of *Chrysomya megacephala*) with the exception of two samples that were positive for *Anaplasma ovis* (MT408585) with identification scores ranging from 99.40–100%. The top hit accession numbers of positive samples in this case were CP037426, CP021120, CP051156, CP050530 and KT827385. In contrast, after sequencing with the *16s rRNA* gene, the positive samples were made up of *Wolbachia* spp. (32) and one uncultured bacterium. For the *16s rRNA* gene, the top hit accession numbers were KX155505, MK184237, MN123078, CP041923, MH447384, AB508951.1 and KT273278. Both positive samples found with *Anaplasma ovis* (MT408585) using the *23s rRNA* gene were identified as *Wolbachia* spp. (AB508951 and KX155505) using the *16s rRNA* gene (Appendix A). Similarity varies from 97.5–100% and 98.2–100% after BLAST query on GenBank using the *16s rRNA* and *23s rRNA* genes, respectively.

Figure 1 shows the resulting *16s rRNA* gene tree. This tree indicates that the majority of *Wolbachia* identified in this study are more closely related to each other than to other known *Wolbachia* strains included as references (Figure 1). Out of 17 sequences included in the phylogenetic analysis, six (35.29%) clustered with Clade B while three (17.64%) clustered with Clade A, and one clustered (5.88%) with Clade T. The remainder (seven sequences, 41.17%) were not well grouped according to the clusters.

## 4. Discussion

In this study, we demonstrated the natural infection of this endosymbiont in field-collected mosquitoes in Cameroon through the combination of qPCR, standard PCR-based *Wolbachia* screening and sequencing (using the *16s rRNA* and *23s rRNA* genes). To the best of our knowledge, *Wolbachia* has thus far only been detected in *A. moucheti* in Cameroon [10]. The successful detection of this bacteria through the *16s rRNA* gene compared to standard detection of the specific gene of *Wolbachia* (*wsp*) has increased the evidence of the presence of this bacteria in *Anopheles* mosquitoes [8,9,10,37,38].

In general, the prevalence of infection was low compared to that which has been reported in other studies in Africa, with 56.6% of *A. moucheti* in Cameroon [10]. Previous studies revealed a prevalence ranging from 3.14–7.75% in Tanzania [37], 28.1% in Singapore [39], 37.1% in Thailand [40] and 26.38% in Sri Lanka [13] for multiple mosquito species. This low infection rate may be due either to environmental differences between areas or to the potential low density of *Wolbachia* in our mosquitoes, as observed elsewhere [41,42]. It is also common to see varying *Wolbachia* infection rates from the same insect host collected at different geographical locations, as in the case of *A. aegypti* from New Mexico (57.4%), Texas (0%) and Florida (4.75%) [43], or the case of *A. demeilloni,* where *Wolbachia* was detected in 38.7% of specimens from Kenya, and 89.3% and 100% in specimens from DRC in 2015 and 2019, respectively [10].

None of the major malaria vectors were found to harbour *Wolbachia*, except the secondary vector *A. paludis* (1/177) collected in Nyabessan in 2016. To our knowledge, this is the first time that *Wolbachia* spp. has been detected in this species in Africa or elsewhere. This low detection of *Wolbachia* in *Anopheles* mosquitoes aligns with other studies [41], strengthening the evidence of their incompatibility with anophelines and revealing possible contamination from environmental sources such as breeding water.

More *Culex* mosquitoes were found to be infected by *Wolbachia*, but the prevalence was low compared to other studies [11,13]. The species screened for bacterial infection in this study included *C. moucheti* (vector of Ntaya virus), *C. quinquefasciatus* (potential vector of bancrofti filariasis and arboviruses such as West Nile, Babanki and Western equine encephalitis viruses), *C. duttoni* (vector of *Arb11266*), *C. perfuscus* (possible vector of Zika virus), *C. univittatus* (vector of West Nile, Usutu, Wesselsbron, Sindbis, Rift valley fever and Spondweni viruses) and *L. tigripes* (vector of Ntaya, Kamese, Mossuril, Sindbis and Babanki viruses). *Wolbachia*-infected species included *C. quinquefasciatus*, *C. moucheti*, *C. perfuscus* and *C. wigglesworthi*. Previous studies reported *Wolbachia* in *C. quinquefasciatus* [11,13,44]. Regarding the other species, this is the first time they have been detected with these bacteria. The bacterial infection of mosquitoes may vary in space and time. In Thailand, *Culex* mosquitoes were not infected by *Wolbachia* [45], while the study by Wiwatanaratnabutr [40] revealed *Wolbachia* infections in *C. tritaeniorhynchus* and *C. gelidus*.

Although previous studies reported the natural infection of *Wolbachia* in both major vectors of dengue and zika, *A. aegypti* and *A. albopictus* [13,43,46], this study reported a low prevalence of infection only in the invasive mosquito *A. albopictus*. In addition, one sample of *A. contigus* was found to be infected. *Wolbachia*-infected *Coquillettidia* mosquitoes were also observed for the first time. 

## 5. Conclusions

This study revealed the natural presence or evidence of *Wolbachia* in field-collected mosquitoes from Cameroon. Albeit at a very low frequency (1.87%), *Wolbachia* was detected in *A. albopictus*, *A. contigus*, *C. moucheti*, *C. quinquefasciatus*, *C. perfuscus*, *C. wigglesworthi* and *Coquillettidia* spp. Future studies are needed to characterise these strains and to determine the impact they might have on disease transmission. The exploration of other areas in order to have an understanding of diverse and consistent mosquito populations should also take place, with the aim of building a greater understanding of species carrying *Wolbachia*, since very few specimens and samples were screened for some mosquito species from this study, which is not representative of the population of the species in question.

## Figures and Tables

**Figure 1 insects-12-01133-f001:**
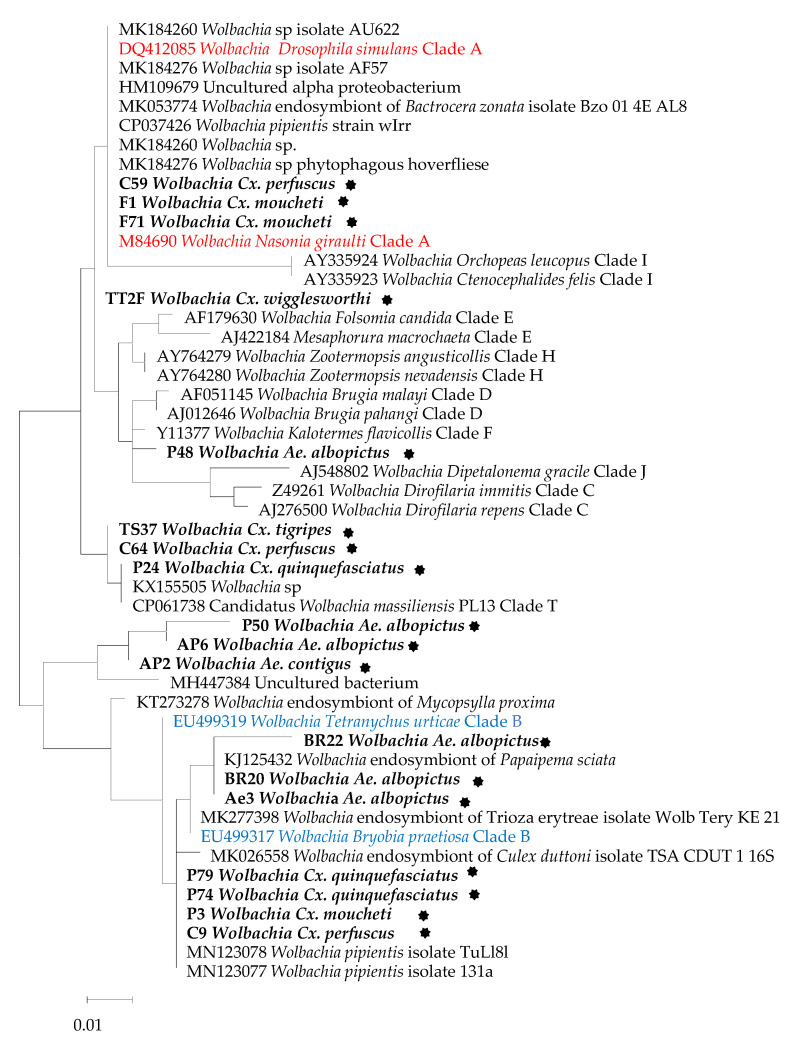
***Wolbachia* strain phylogenetic analysis using the *16S rRNA* gene.** The evolutionary history was inferred using the maximum likelihood method and Tamura–Nei model [35]. The tree with the highest log likelihood (−1118,14) is shown. The initial tree(s) for the heuristic search were obtained automatically by applying the neighbour-joining and BioNJ algorithms to a matrix of pairwise distances estimated using the Tamura–Nei model, and then by selecting the topology with the higher log likelihood value. The tree is drawn to scale, with branch lengths measured in the number of substitutions per site. This analysis involved 61 nucleotide sequences. There were a total of 350 positions in the final [13] dataset. Evolutionary analyses were conducted in MEGA11 [36]. Sequences from this study are indicated in bold, while the two main groups, Clade A (red) and Clade B (blue), are coloured. The *Wolbachia* strains obtained in this study are in bold and marked with a black star.

**Table 1 insects-12-01133-t001:** Primers and probes used for real-time quantitative and standard PCR in this study.

Microorganisms	Targeted Sequence		Primers F, R (5′-3′) and Probes p (6FAM-TAMRA)
qPCR primers			
*Anaplasmataceae*	*23S*	TtAna_F	TGACAGCGTACCTTTTGCAT
		TtAna_R	GTAACAGGTTCGGTCCTCCA
		TtAna_P	6FAM-GGATTAGACCCGAAACCAAG
*Bartonella*	*ITS*	Barto_ITS2_F	GGGGCCGTAGCTCAGCTG
		Barto_ITS2_R	TGAATATATCTTCTCTTCACAATTTC
		Barto_ITS2_P	6FAM-CGATCCCGTCCGGCTCCACCA
*Borrelia*	*16S*	Bor_16S_3F	AGCCTTTAAAGCTTCGCTTGTAG
		Bor_16S_3R	GCCTCCCGTAGGAGTCTGG
		Bor_16S_3P	6FAM-CCGGCCTGAGAGGGTGAACGG
	*ITS4*	Bor_ITS4_F	GGCTTCGGGTCTACCACATCTA
		Bor_ITS4_R	CCGGGAGGGGAGTGAAATAG
		Bor_ITS4_P	6FAM-TGCAAAAGGCACGCCATCACC
*Coxiella burnetii*	*IS1111A*	CB_IS1111_0706F	CAAGAAACGTATCGCTGTGGC
		CB_IS1111_0706R	CACAGAGCCACCGTATGAATC
		CB_IS1111_0706P	6FAM-CCGAGTTCGAAACAATGAGGGCTG
	*Hyp. Protein* *IS30A*	CB_IS30A_3F	CGCTGACCTACAGAAATATGTCC
		CB_IS30A_3R	GGGGTAAGTAAATAATACCTTCTGG
		CB_IS30A_3P	6FAM-CATGAAGCGATTTATCAATACGTGTATGC
*Rickettsia* spp.	*gltA*	RKND03_F	GTGAATGAAAGATTACACTATTTAT
		RKND03_R	GTATCTTAGCAATCATTCTAATAGC
		RKND03 P	6FAM-CTATTATGCTTGCGGCTGTCGGTTC
Standard PCR primers			
Invertebrate identification (Folmer)	*COI*	LCO1490	GGTCAACAAATCATAAAGATATTGG
		HCO2198	TAAACTTCAGGGTGACCAAAAAATCA
*Anaplasmataceae*	*23S*	Ana23S-212f	ATAAGCTGCGGGGAATTGTC
		Ana23S-753r	TGCAAAAGGTACGCTGTCAC
*Wolbachia*	*16S*	W-SpecF	CATACCTATTCGAAGGGATAG
		W-SpecR	AGCTTCGAGTGAA ACCAATTC

**Table 2 insects-12-01133-t002:** Prevalence of *Wolbachia* spp. detected in mosquitoes collected in Cameroon.

Species	Tested	*Wolbachia* Positive (%)
*Anopheles gambiae* s.l.	171	0 (0)
*Anopheles paludis*	211	1 (0.4%)
*Anopheles nili*	2	0 (0)
*Anopheles moucheti*	20	0 (0)
*Culex (Lutzia) tigripes*	18	1 (5.55)
*Culex duttoni*	43	0 (0)
*Culex perfuscus*	116	4 (3.44)
*Culex quinquefasciatus*	730	7 (0.95)
*Culex univittatus*	11	0 (0)
*Culex moucheti*	15	7 (46.67)
*Culex wigglesworthi*	1	1 (100)
*Culex* sp.	52	2 (3.84)
*Aedes albopictus*	155	8 (5.16)
*Aedes aegypti*	43	0 (0)
*Aedes africanus*	4	0 (0)
*Aedes simpsoni*	6	0 (0)
*Aedes contigus*	32	1 (3.12)
*Aedes* sp.	6	0 (0)
*Coquillettidia* spp.	7	1 (14.28)
*Mansonia africana*	42	0 (0)
*Mansonia uniformis*	55	0 (0)
Total	1740	33 (1.89)

## Data Availability

All the data from the study is available in the manuscript.

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
