# Peer review of "Wolbachia Detection in Field-Collected Mosquitoes from Cameroon"

_insects, 2021, doi:10.3390/insects12121133_

Round 1
Reviewer 1 Report
These authors main work in this study is to detect Wolbachia in mosquitoes from the Cameroon. The results show that the presence of Wolbachia in the mosquitoes are very low. The methods for the detection of Wolbachia should be mentioned in brief and not just provide a reference. The method cannot be followed by others readers. All primers and probes used must be listed
Study by Ref( 8) shows that the Wolbachia is present in large numbers of the mosquitoes but not the present study. Thus I feel that the authors should try another method for the detection of Wolbachia and see if they get similar results. What puzzles me is that even in Ae. albopictus the presence of Wolbachia is very very low. Reasons must be discussed.
I would trust Ref 8 because the authors have done insitu hybridisation on ovaries to prove the presence of Wolbachia.
Thus please try nested PCR and other methods to see if Wolbachia is present in the ovaries.
Limitations of this study should also be discussed in greater detail.
Author Response
The Editor in chief
Insects Marseille, 30 November 2021
Dear Sir/Madam,
RE: RESUBMISSION OF MS# insects-1471016
I will like to thank the reviewers for their comments and suggestions to improve the quality of the manuscript. All requested changes were done accordingly and are highlighted in the main text.
Attached herewith please find point to point response to the reviewers comments and suggestion which have been highlighted in the manuscript as underlined.
I hope these responses addresses the concerns raised.
Yours Sincerely,
Reviewer #1
point 1: These authors main work in this study is to detect Wolbachia in mosquitoes from the Cameroon. The results show that the presence of Wolbachia in the mosquitoes are very low. The methods for the detection of Wolbachia should be mentioned in brief and not just provide a reference. The method cannot be followed by others readers. All primers and probes used must be listed
Response 1: We thank the reviewer for this remark, the manuscript was revised and all methods described as requested. Methodology for mosquito collection/sampling and detection of Wolbachia or other bacteria was amended, Please see lines 91-102 as follow: CDC-LTs collections were conducted both indoor and outdoor in selected houses (04) from 1900 hrs to 0600 hrs during at least 3 consecutive nights per season. Traps were placed indoors near someone sleeping under a net at about 1.5m from the ground and outdoor in the veranda. The same number of houses were selected randomly per col-lection site for HLCs. In each houses, indoor and outdoor mosquito collections were carried out from 1900 hrs to 0600 hrs by 2 teams of 2 people per house using mouth aspirators. Outdoor mosquito collection was carried out about 10 m from each house se-lected for indoor mosquito collection by HLCs. The 2 teams (indoor/outdoor collectors) exchanged positions in each homestead every hour of the night. Collection bags for mosquito collection were exchanged hourly following mosquito capture. Immature stages in breeding sites (abandoned tyres, metallic and plastic containers, gutters, stagnant water pools,) were collected using dipping technique while sweep nets were used to catch mosquito adults resting on vegetation.
In addition, we included all primers and probes used during this study (see Table 1). For bacteria detection, the first analysis was done using pooled DNA mosquitoes and individual mosquitoes from positive pools were then analyzed using qPCR. Positive individual were sequenced using 16s rRNA Wolbachia and 23s rRNA Anaplasmataceae Please see lines 140-144 as follow: Mosquitoes were pooled (4-10 individual mosquitoes) and screened for the detection of Rickettsia spp., Borrelia spp., Bartonella spp., Coxiella burnettii and Anaplasmataceae using quantitative PCR (qPCR) [32,33]. In case a pool was found positive for one bacte-ria, specimens were then analysed individually using the same method.
Point 2: Study by Ref ( 8) shows that the Wolbachia is present in large numbers of the mosquitoes but not the present study. Thus I feel that the authors should try another method for the detection of Wolbachia and see if they get similar results. What puzzles me is that even in Ae. albopictus the presence of Wolbachia is very very low. Reasons must be discussed. I would trust Ref 8 because the authors have done insitu hybridisation on ovaries to prove the presence of Wolbachia.
Thus please try nested PCR and other methods to see if Wolbachia is present in the ovaries.
Response 2: We thank the reviewer for this suggestion. Yes, the study by ref (8) where I co-authored showed high prevalence of Wolbachia but only in An. moucheti and An. demeilloni. The prevalence of Wolbachia in other mosquito species analyzed during the study was low and species dependent (Master thesis report at LSHTM).
We all agree that the prevalence of Wolbachia is very high in some mosquitoes like Aedes albopictus and Culex pipiens sl. It is also important to note that the variation of the prevalence of a pathogen, Wolbachia in this case, depends on several factors such as environmental factors (temperature, geographic location and mosquito-borne diseases leading to its variation in space and time. In our case, mosquitoes were collected in Cameroon (mostly during the dry season), which is a hot zone with a large circulation of malaria and arboviruses (Bamou et al 2021 https://doi.org/10.1186/s13071-021-04950-9), which may be the cause of the observed low prevalence. In aa study conducted in Vietnam, the authors showed a reduction in the prevalence of Wolbachia in trans-infected Aedes aegypti according to climatic conditions and areas (https://doi.org/10.12688/gatesopenres.13347.1). Another study done in China showed a high prevalence that varied according to localities (https://doi.org/10.1186/s13071-020-3899-4).
Concerning our study, the 16s RNA genes was used compared to wsp and coxA for ref (8). We think that this low prevalence may be due to the low density of Wolbachia in mosquito populations from Cameroon.
In addition, comparison of 16sRNA amplified using regular PCR (W16S-Spec) (25), nested PCR (W16S-WE), and a qPCR assay by Gomes et al (2017) showed that the qPCR method was more sensitive for detecting Wolbachia (W+) in 67% of mosquitoes with high correlation between technical replicates. Whereas regular PCR was the least sensitive method and detected Wolbachia in 16% of the samples (11/69) while nested PCR found 52% of samples (36/69) positive. Please find enclosed the link of that publication (https://doi.org/10.1073/pnas.1716181114)
In our laboratory, we do not use nested PCR because can increase the risk of contamination.
Point 3: Limitations of this study should also be discussed in greater detail.
Response 3: done. We include this part in the conclusion of the manuscript. See line 253-259 “In addition, further investigation are needed to identify factors that could shaping their prevalence on some particular mosquitoes species. Also, exploring more areas to have diversify and consistent mosquitoes populations should be done, to have a deep understanding species carrying Wolbachia; since for some species only a few mosquitoes were tested (n < 10), which is not enough to represent the whole population of that species”.

Reviewer 2 Report
The manuscript contains new finding on Wolbachia bacteria. However, there are several weak points, especially methodology that needs to redo. Also, several typos have been found (see attached files).

Author Response
The Editor in chief
Insects Marseille, 30 November 2021
Dear Sir/Madam,
RE: RESUBMISSION OF MS# insects-1471016
I will like to thank the reviewers for their comments and suggestions to improve the quality of the manuscript. All requested changes were done accordingly and are highlighted in the main text.
Attached herewith please find point to point response to the reviewers comments and suggestion which have been highlighted in the manuscript as underlined.
I hope these responses addresses the concerns raised.
Yours Sincerely,
Reviewer #2
All typos and grammatical error were corrected and are underlined in the main text.
Abstract
Point 1: “Phylogenetic analysis of the 16S RNA sequences …subgroups (A and B)” need revision, not clear
Response 1: We thank the reviewer for this observation, the sentence was revised for clarification as follow “Phylogenetic analysis of the 16S RNA sequences show they belong mainly to two distinct subgroups (A and B)”. Please see line 34.
Introduction
Point 2: abbreviations (LLIN, WHO, IRS) should be delate because mentioned once.
Response 2: done
Point 3: It has long been said that Wolbachia spp. were absent in some mosquitoes such as Ae. aegypti and Anopheles mosquitoes. Which species of Anopheles?
Response 3: We thank the reviewer for this question. These Anopheles species include the main major malaria vector Anopheles gambiae.
Query 4: Explain how Wolbachia change the biology of mosquito
Response 4: Wolbachia could act by reducing fertility of mosquito/host, or by impacting replication parasite or blocking virus/transmission in their host. It can interact in many various ways with their hosts by changing their biology by cytoplasmic incompatibility. https://doi.org/10.1016/j.ibmb.2004.03.025
Materials and methods
Point 5: Biological samples. This section need more details for each collection method.
Response 5: Thank for this remark, we provided more clarification in that part as follow. Please see lines 87-99. The four (4) mosquito collection methods were described in the reversed version giving the number of house used for CDC LT and HLC. “CDC-LTs collections were conducted both indoor and outdoor in selected houses (04) from 1900 hrs to 0600 hrs during at least 3 consecutive nights per season. Traps were placed indoors near someone sleeping under a net at about 1.5m from the ground and outdoor in the veranda. The same number of houses were selected randomly per collection site for HLCs. In each houses, indoor and outdoor mosquito collections were carried out from 1900 hrs to 0600 hrs by 2 teams of 2 people per house using mouth aspirators. Outdoor mosquito collection was carried out about 10 m from each house selected for indoor mosquito collection by HLCs. The 2 teams (indoor/outdoor collectors) exchanged positions in each homestead every hour of the night. Collection bags for mosquito collection were exchanged hourly following mosquito capture. Immature stages in breeding sites (abandoned tyres, metallic and plastic containers, gutters, stagnant water pools,) were collected using dipping technique while sweep nets were used to catch mosquito adults resting on vegetation”.
Point 6: Which species used for identification confirmation by MALDI TOF MS and molecular biology
Response 6: MALDI TOF MS (>95% of specificity) was used to identify all mosquitoes (having their legs) in this study by crosswise query against the in-house database available at IHU Mediterranée Infection. For species absent in this database, five MS spectra reproducible with good intensity were introduced in the database for crosswise query. Molecular identification of these specimens were generated before introduction of their MS spectra in the database. In addition, species positive to Wolbachia were sequenced.
Point 7: removed Bamou et al unpublished.
Response 7: done.
Point 8: Absence of other bacteria in the Table 1. Consistency of spelling
Response 8: During our study, we did not find any mosquito with other bacteria than Wolbachia, reason why we chose to remove columns with Borelia, Coxiella burneti and Ricketsia in the Table.
All the species names were harmonized in the text and table as requested.
Point 9: Figure to amend
Response 9: figure was amended and references corrected in figure legend
Point 10: References need careful revision
Response 10: References were reviewed and corrected accordingly to journal style

Reviewer 3 Report
The authors tested for the presence of Wolbachia and four other bacterial groups in mosquitoes collected in Cameroon. The impact and motivation of the paper are not well-articulated and must be sharpened. Some key details are missing from the Materials and Methods and Results sections that are important for a clear understanding of the work done.
More comments are in the attached pdf.

Author Response
The Editor in chief
Insects Marseille, 30 November 2021
Dear Sir/Madam,
RE: RESUBMISSION OF MS# insects-1471016
I will like to thank the reviewers for their comments and suggestions to improve the quality of the manuscript. All requested changes were done accordingly and are highlighted in the main text.
Attached herewith please find point to point response to the reviewers comments and suggestion which have been highlighted in the manuscript as underlined.
I hope these responses addresses the concerns raised.
Yours Sincerely,
Reviewer #3
Point 1: Some editing is required throughout the text due to spelling errors, italicization, awkward word choice and sentence structure, etc. A few examples are listed below:
- Introduction. “Distribution of Wolbachia could be occur(?) horizontally between
mosquito populations…”
- Materials and Methods. “Mosquitoes were stored either in silicate or in empty
containers (Eppendorf tubes or petri dishes without a conservator product) at room temperature, or in silicate, RNAlater, 70% alcohol and in empty containers in the fridge at -20°C, for two to 48 months depending on the time of collection.”
Simple summary. “Wolbachia bacteria from different strains, carried by many insects and nematodes can interact in many various ways with their hosts by changing their biology in different manners (including?) through the suppression of vector population and reduction of parasite transmission”.
Wolbachia changes host biology in many ways and the way the above sentence is worded suggests these changes are limited to only suppression of vector population and reduction of parasite transmission, ignoring the fact that before being known as a popular method of vector control, Wolbachia was already extensively studied for causing drastic changes in host reproduction biology, as described in Reference no. 12.
Response 1: We thank the reviewer for these remarks. All these observation have corrected in the manuscript accordingly.
Point 2: Introduction. “Distribution of Wolbachia could be horizontally between mosquito populations”. The authors should provide a reference for this as Reference no.8 demonstrated only vertical transmission of Wolbachia.
Response 2: We thanks the reviewer for this observation, references were added to show the horizontal transmission of Wolbachia in insects.(doi:10.1186/s12862-016-0660-x, doi:10.3389/fmicb.2017.02237).
Point 3: Introduction. “Although there are many studies on the use of Wolbachia for mosquito control, their prevalence in wild collected mosquito remains limited.” Is it their prevalence that remains limited? Or knowledge about their prevalence? Wolbachia is highly prevalent in species such as Aedes albopictus (wAlbA and wAlbB) [https://doi.org/10.1186/s13071-020-3899-4] and Culex pipiens (wPip) [https://doi.org/10.1186/s13071-018-2777-9].
Response 3: Dear reviewer, thanks for the comments. We all agree that the prevalence of Wolbachia is very high in some mosquitoes like Aedes albopictus and Culex pipiens sl. It is also important to note that the variation in the prevalence of a pathogen, Wolbachia in this case, would depend on several factors such as environmental factors (temperature, geographic location and mosquito-borne diseases) leading to it variation in space and time. In our case, mosquitoes were collected in Cameroon (during the dry season for the most part), which is a hot zone with a large circulation of malaria and arboviruses (Bamou et al 2021 https://doi.org/10.1186/s13071-021-04950-9), which could possibly be the cause of this low prevalence.
For example, a study conducted in Vietnam showed a reduction in the prevalence of Wolbachia in trans-infected Aedes aegypti depending on climatic conditions and area (https://doi.org/10.12688/gatesopenres.13347.1). In addition, another study done in China shows that despite the prevalence is high, it varies according to localities (https://doi.org/10.1186/s13071-020-3899-4).
Thanks for pointing this. It is the knowledge about the prevalence of Wolbachia that remains limited.
For the continuity of this work, other primers will be used.
Point 4: To a Wolbachia researcher, the Introduction leaves the reader with the impression that the authors are not up to date with the state-of-the-art on Wolbachia research. They seem to be aware that Wolbachia is used as a biological control strategy but cannot pinpoint clearly how the strategy functions. Thus, to non-Wolbachia researchers, the importance of the endobacterium and of this study is not obvious.
Wolbachia-infected mosquitoes are used in several countries in population replacement or population suppression approaches, yet the authors cite “for future use in biological control development using the para-transgenesis approach”, adding to the confusion. It is not clear how knowledge of Wolbachia prevalence in Cameroon mosquitoes could lead to the development of Wolbachia as a paratransgenesis approach for biological control.
The authors may benefit from this literature:
https://doi.org/10.1038/s41579-018-0025-0, https://doi.org/10.1016/j.pt.2021.06.007
Response 4: We thank the reviewer for this comment. There is not enough data of Wolbachia in mosquitoes from Cameroon and this study provides an opportunity to determine novel Wolbachia strains that have not yet been characterised. Novel Wolbachia strains are interesting candidates for vector control as they may confer useful phenotypes when transinfected into alternative mosquito hosts.
The manuscript was amended for better understanding. Although this bacteria was highly prevalent (56%) in malaria vectors An. moucheti from Cameroon, their distribution among others mosquito groups remain poorly under-stood [10]. In addition, the knowledge about their prevalence remains limited. There is not enough data of Wolbachia in mosquitoes from Cameroon and this study provides an opportunity to determine novel Wolbachia strains that have not yet been characterised. Novel Wolbachia strains are interesting candidates for vector control as they may con-fer useful phenotypes when transinfected into alternative mosquito hosts. In the pre-sent study, current prevalence of Wolbachia on different mosquitoes species was de-termined and subsequently, their sequences and phylogenetic were also investigated using 23S rRNA Anaplasmataceae and Wolbachia 16S rRNA targeting 550bp and 460bp of 16S and 23S genes respectively.
Point 5: Materials and methods. It is not clear how the protein mass spectra of mosquito samples can allow for species identification. Is it compared to a reference database of known protein spectra? Although the authors cite their own unpublished manuscript (which is not normal), this minimal detail should be explained.
Response 5: The manuscript was amended. Yes, the Mass Spectra protein profiles of mosquito sample are compared to in-house database with known mosquito species MS spectra. After comparison, the program gives a score (log score value) comprised between 0-3. If the LSV is greater than the threshold (1.8), identification is correct. As shown in these publications:
https://doi.org/10.1186/s13071-020-04234-8; https://doi.org/10.4269/ajtmh.20-0031
https://doi.org/10.1371/journal. pone.0234098, doi: 10.2217/fmb-2020-0145.
We modified the text as follow “Mosquito species identifications were confirmed by MALDI TOF MS and molecu-lar biology analysis and results are presented elsewhere. Briefly, specimen’ legs were introduced in 1.5ml Eppendorf tubes containing 20µL of protein extraction mix (70% formic acid (v/v) and 50% acetonitrile (v/v)) and grinded using glass beads in Tissue Lyser [29,30]. One microliter of each protein extract was deposited in target plate in quadruplicate and one microliter of matrix was spotted on top of each spot and al-lowed to dry. Protein mass spectra were obtained with a Microflex LT MALDI-TOF mass spectrometer (Bruker Daltonics, Germany) [31] and crosswise tested against in-house database available at VITROME containing spectra of known mosquito species. Results are given by the reliability of species identification, estimated as the log score value (LSV) and calculated using a biostatistical algorithm from the MALDI Biotyper software v.3.0. The LSV range from 0-3 and value equals or greater than 1.8 is consid-ered as threshold for species identification of mosquitoes.”.
Point 6: Materials and methods. Further details should be added to understand the methods and why certain analyses were performed:
Response 6: Done
Mosquitoes were pooled (4-10 individual mosquitoes) and screened for the detection of Rickettsia spp., Borrelia spp., Bartonella spp., Coxiella burnettii and Anaplasmataceae using quantitative PCR (qPCR) [32,33]. In case a pool was found positive for one bacteria, specimens were then analysed individually using the same method. Target genes, primers and probes sequences (Table 1) were used as previously described in other studies [32,33]. DNA from our laboratory-cultured strains of Borrelia crocidurae, Rickettsia africae, Bartonella quintana, Coxiella burnetii and Ehrlichia canis were used as positive controls. DNA free of pathogens from laboratory reared ticks was used as a negative control. Samples were considered positive with Ct < 36 for all bacteria tested. Later on, samples (from individual specimens) positive for Anaplasmataceae underwent 23s RNA and 16s RNA genes standard PCR amplification and sequencing as described by Diarra et al.[32] .
Point 7: Were the mosquitoes used for DNA extraction individually or in pools?
Response 7: DNA was extracted from individual mosquitoes but the first qPCR was performed on pooled DNA (4-10 individuals) mosquitoes and when positive, the individual DNA of the same pool were analysed using the same methodology. Positive individuals were sequenced after. Please see lines 130-133.
Point 8: The authors shouldn’t reference methodology from an unpublished manuscript as there is no way for the reviewers or the readers to evaluate the quality and correctness of the methodology used. Those methodologies should be described in full.
Response 8: We thank the reviewer for giving us the opportunity to clarify this issue. the Methodology was amended and described in detail. https://doi.org/10.2217/fmb-2020-0145
Point 9: The target genes for Mosquito microbiota analysis should at least be listed in this
article. Are they different from Anaplasmataceae genes (23S RNA and 16s RNA genes)?
Response 9: We thank the reviewer for this question. The primers used during this study are listed in Table 1. 23S rRNA and 16s rRNA genes were used only for Anaplasmataceae and Wolbachia detection. For other primers, see Table 1.
Point 10: Was sequencing only done for Anaplasmataceae-positive samples? Why?
Response 10: Yes, only anaplasmataceaa-positive sample were sequenced using both 23s and 16s rRNA genes because the gene is not specific, as it amplifies Wolbachia, Erlichia and Anaplasma gene fragment
Point 11: What were the parameters used for the BLAST and multi-alignment analysis? If default parameters were used, this should be stated.
Response 11: Yes, the parameters used for the BLAST and multi-alignment analysis were the default parameters and we have added this to the manuscript as follows: “The resulting sequences were used to perform BLAST searches via the National Center for Biotechnology Information (NCBI) GenBank database and then aligned using BioEdit using the proposed software default parameters”.
Point 12: The Materials and Methods say the phylogenetic tree was constructed using MEGA7 but the Figure 1 legend says MEGA11.
Response 12: Thanks for this observation. The sentence was corrected. Please see line 160
Point 13: Why does this study choose to focus on the five bacterial groups Rickettsia spp., Borrelia spp., Bartonella spp., Coxiella burnettii and Anaplasmataceae? Is it because these are the only mosquito-associated bacteria in Cameroon? Is there a reference for that?
Response 13: Results from this manuscript were part of a large project on “Development of proteomic tools for identification of mosquitoes (and other arthropods) from Cameroon and their associated pathogens”. In this short manuscript, we focused only on bacteria that are mainly studied in our laboratory and are considered as emergent mosquito-borne or vector-borne pathogens (Maina et al 2017; Zhang et al 2019;Maureen et al 2016; Dieme et al 2015).
https://doi.org/10.1371/journal.pone.0188327; https://doi.org/10.1089/vbz.2019.2456; https://doi.org/10.1128/microbiolspec.MTBP-0017-2016 , https://doi.org/10.1073/pnas.1413835112
Point 14: Results. Which Wolbachia strains were found in which mosquito samples?
Response 14: Here, we did not characterized our Wolbachia strain. Using phylogenic analysis, we observed that most of the Wolbachia positive samples were close to clade A and B. For specimens close to clade A, we have positive samples from Cx. perfuscus, Cx. moucheti, Cx. wigglesworthi, Cx. tigripes, Cx quinquefasciatus and Ae. albopictus. For specimens close to Clade B, positive sample from Ae. albopictus, Cx. perfuscus, Cx. quinquefasciatus and Cx. moucheti were found. Others were grouped with Clade T as found in Figure 1.
Future study will include characterization of the strain found using different target genes.
Point 15: Figure 1. Which are the Wolbachia strains identified in this study?
Response 15: In this study, sample in red are identified in this study.Using phylogenic analysis, we observed that most of the Wolbachia positive samples were close to clade A and B. For specimens close to clade A, we have positive samples from Cx. perfuscus, Cx. moucheti, Cx. wigglesworthi, Cx. tigripes, Cx quinquefasciatus and Ae. albopictus. For specimens close to Clade B, positive sample from Ae. albopictus, Cx. perfuscus, Cx. quinquefasciatus and Cx. moucheti were found. Others were grouped with Clade T as found in Figure 1.
Considering BLAST results, we can say that strain obtained were close to WNik, Wirr, wMeg, WbHpNm using 23s rRNA.
Future studies will include characterization of the strain found using different target genes.
Point 16: Results. What are the nucleotide similarity or BLAST results for the sequenced samples?
Response 16: The similarity varied from 97.5-100% and 98.2-100% after BLAST query on GenBank using 16sRNA and 23s rRNA respectively.
Point 17: Discussion. Was the Wolbachia screening qPCR-based? Or PCR-based?
Response: We thanks the reviewer for this question. Both methods were used as described in the text. DNA was extracted from individual mosquitoes but the first qPCR was performed in pool of mosquitoes and when positive, the individual DNA of the same pool were analysed using the same methodology. Positive individuals were sequenced later.
We have clarified the sentence as follows “In this study, we demonstrated the natural infection of field-collected mosquitoes in Cameroon by Wolbachia sp. using a combination of qPCR, standard PCR and sequencing (using 16s rRNA and 23s RNA genes)”.
The Materials and methods section suggests the latter.
Point 18: Discussion. What is the reasoning for using rRNA genes instead of the wsp gene?
Response 18: Dear reviewer, based on previous studies done in Malaysia, we used of rRNA genes instead of wsp. Results of these study by Wong et al (2020) showed that rRNA genes (16srRNA) was more efficient in the detection of Wolbachia in mosquito than wsp genes. Please see the DOI of that paper for more clarification
https://doi.org/10.1186/s13071-020-04277-x
In addition, comparison of 16sRNA amplified using regular PCR (W16S-Spec), nested PCR (W16S-WE), and a qPCR assay by Gomes et al (2017) showed that the qPCR method was more sensitive detecting Wolbachia (W+) in 67% of mosquitoes with high correlation between technical replicates. While regular PCR was the least sensitive method and detected Wolbachia in 16% of the samples (11/69) while nested PCR found 52% of samples (36/69) positive. https://doi.org/10.1073/pnas.1716181114
In our laboratory, the 16s rRNA gene is always used.
Point 19: Discussion. How different are the infection rates reported in this study compared to other studies (references no. 32,33,11,34,35)?
Response 19: Done. The text was amended as follow. In general, the prevalence of infection was low compared to what has been re-ported in other studies in Africa with 56.6% in Cameroon on An. moucheti [8] and be-tween 3.14-7.75% on An. arabiensis in Tanzania [34] and oriental regions with 28.1% in Singapore [36] 37.1% in Thailand [37] and 26.38% in Sri Lanka [11]. For studies out of Africa, Wolbachia were found in different species of mosquito. Please see lines 211-213
Point 20: When comparing your results to those from other studies, it is normal to explain what the other studies’ results are, instead of leaving your readers with the burden of searching.
Response 20: We thank the reviewer for giving the opportunity to clarify this issue. This has been corrected in the main text
Point 21: Discussion. “In Thailland, Culex mosquitoes were not found to be infected by Wolbachia [11] while Wiwatanaratnabutr [34] has reported Wolbachia infections in Cx. tritaeniorhynchus and Cx. gelidus.” Reference no. 11 is about Sri Lanka, not Thailand. Also, as the authors are comparing Ref 11 to Ref 34, does the study in Ref 11 include Culex tritaeniorhynchus and C.gelidus?
Response 21: Thanks for this point. We did a mistake with the first reference cited in this sentence. Both studies were from the same author and in the same locality but done in different moment. The text was amended correctly as In Thailland, Culex mosquitoes were not found infected by Wolbachia [44] while the study of Wiwatanaratnabutr [39] has reported Wolbachia infections in Cx. tritaeniorhynchus and Cx. gelidus
Point 22: Discussion. “Although natural infection of Wolbachia has been reported in both majorvectors of dengue and zika, Ae. aegypti and Ae. albopictus [11,38,40], this study reported alow prevalence of infection only in the invasive mosquito Ae. albopictus.” It should bespecified that this sentence only concerns Aedes mosquitoes.
Response 22: Done
Point 23: The authors should acknowledge that future studies should investigate the same research question with larger samples numbers, since for some species only a few mosquitoes were tested (n < 10), which is not enough to represent the whole population of that species.
Response 23: Done. Please see conclusion “In addition, further investigation are needed to identify factors that could shaping their prevalence on some particular mosquitoes species. Also, exploring more areas to have diversify and consistent mosquitoes populations should be done, to have a deep understanding species carrying Wolbachia; since for some species only a few mosquitoes were tested (n < 10), which is not enough to represent the whole population of that species.”
Point 24: Author names (prenom, puis nom) and initials under Authors’ contributions should be checked. The contributions of BDT and CAN are not listed
Response 24: Done

Round 2
Reviewer 1 Report
Can be accepted. Authors have made the necessary corrections
Author Response
Dear Reviewer, thanks for your comments. T
The entire manuscript was edited for English language by a specialist as you can found attached.

Reviewer 2 Report
Dear Authors
This study is very impressive and valuable. However, the manuscript needs extensive editing of English language before publishing.
Author Response
Dear Reviewer, thanks you for your comment about the english quality of the Mansucript. It was proofread and corrected by an experienced professional editor. Grammar, spelling, punctuation, sentence structure and phrasing were corrected. The editor is a native English speaker, UK university graduate, and full member of the Institute of Translation and Interpreting, the Chartered Institute of Linguists, and the Société Française des Traducteurs

Reviewer 3 Report
The authors have earnestly made great efforts to address my comments and concerns and the manuscript has notably benefitted from it.
I only have a few more minor comments:
- Line 75-76: "this study provides an opportunity to determine novel Wolbachia strains that have not yet been characterised." While it is true that the authors seem well-placed to do this, the study described in this particular manuscript did not actually characterise Wolbachia strains (see Response 14 from Authors). The Authors explained that characterisation of the strain will instead be done in another study. This phrase then seems misleading.
- Line 235 Usutu, not Usuntu
- Text-editing is particularly needed for Conclusion section (lines 250-259) and while I am flattered that the authors liked the sentence from my comments ("since for some species only a few mosquitoes were tested (n < 10), which is not enough to represent the whole population of that species") enough to put it in their own manuscript, I would suggest not to make a habit of copy-pasting reviewer comments in their future publications.
Author Response
The Editor in chief
Insects Marseille, 6 December 2021
Dear Sir/Madam,
RE: RESUBMISSION OF MS# insects-1471016
I will like to thank the reviewers for their comments and suggestions to improve the quality of the manuscript. All requested changes were done accordingly and are highlighted in the main text.
Attached herewith please find point to point response to the reviewers comments and suggestion which have been highlighted in the manuscript as in yellow. The manuscript was edited by native english speaker, specialist in the translation.
I hope these responses addresses the concerns raised.
Yours Sincerely,
Reviewer #3
I only have a few more minor comments:
Point 1: Line 75-76: "this study provides an opportunity to determine novel Wolbachia strains that have not yet been characterised." While it is true that the authors seem well-placed to do this, the study described in this particular manuscript did not actually characterise Wolbachia strains (see Response 14 from Authors). The Authors explained that characterisation of the strain will instead be done in another study. This phrase then seems misleading.
Response 1 : Thanks for this observation. This was a short communication about what we found in our study and we intend to use others genes for the characterisation of obtained Wolbachia.
The text was amended and highlighted in yellow in the as follow ” In addition, knowledge about their prevalence remains limited. Data regarding Wolbachia in mosquitoes in Cameroon is lacking and this study provides an opportunity to identify new Wolbachia strains that could be characterised”.
Point 2 :Line 235 Usutu, not Usuntu
Response 2 : Dear reviewer, thanks for this observation. The virus name was corrected in the main text.
Point 3: Text-editing is particularly needed for Conclusion section (lines 250-259) and while I am flattered that the authors liked the sentence from my comments ("since for some species only a few mosquitoes were tested (n < 10), which is not enough to represent the whole population of that species") enough to put it in their own manuscript, I would suggest not to make a habit of copy-pasting reviewer comments in their future publications.
Response 3: Thanks for this observation.
The text was modified a bit as follow ” This study revealed the natural presence or evidence of Wolbachia in field-collected mosquitoes from Cameroon. Albeit at a very low frequency (1.87%), Wolbachia was detected in Ae. albopictus, Ae. contigus, Cx. moucheti, Cx. quinquefasciatus, Cx. per-fuscus, Cx. wigglesworthi and Coquillettidia spp. Future studies are needed to characterise these strains and to determine the impact they might have on disease transmission and their potential role in biological control. In addition, further investigations are needed to identify factors that could shape their prevalence on particular mosquitoes species. The exploration of other areas in order to have an understanding of diverse and consistent mosquito populations should also take place, with a view to having a greater understanding of species carrying Wolbachia, since very few specimens/samples were screened for some mosquito species from this study, which is not representative of the population of the species in question.”.
